# Evaluating Sex Steroid Hormone Neuroprotection in Spinal Cord Injury in Animal Models: Is It Promising in the Clinic?

**DOI:** 10.3390/biomedicines12071478

**Published:** 2024-07-04

**Authors:** Angélica Coyoy-Salgado, Julia Segura-Uribe, Hermelinda Salgado-Ceballos, Tzayaka Castillo-Mendieta, Stephanie Sánchez-Torres, Ximena Freyermuth-Trujillo, Carlos Orozco-Barrios, Sandra Orozco-Suarez, Iris Feria-Romero, Rodolfo Pinto-Almazán, Gabriela Moralí de la Brena, Christian Guerra-Araiza

**Affiliations:** 1CONAHCyT-Unidad de Investigación Médica en Enfermedades Neurológicas, Hospital de Especialidades Dr. Bernardo Sepúlveda, Centro Médico Nacional Siglo XXI, Instituto Mexicano del Seguro Social, Mexico City 06720, Mexico; crls2878@gmail.com; 2Subdirección de Gestión de la Investigación, Hospital Infantil de México Federico Gómez, Secretaría de Salud, Mexico City 06720, Mexico; jujeseur@gmail.com; 3Unidad de Investigación Médica en Enfermedades Neurológicas, Hospital de Especialidades Dr. Bernardo Sepúlveda, Centro Médico Nacional Siglo XXI, Instituto Mexicano del Seguro Social, Mexico City 06720, Mexico; melisalce@yahoo.com (H.S.-C.); tzayakita@gmail.com (T.C.-M.); stephanie.sanchez.torres@gmail.com (S.S.-T.); sorozco5@hotmail.com (S.O.-S.);; 4Sección de Estudios de Posgrado e Investigación, Escuela Superior de Medicina, Instituto Politécnico Nacional, Plan de San Luis y Díaz Mirón, Mexico City 11340, Mexico; 5Unidad de Investigación Médica en Farmacología, Hospital de Especialidades Dr. Bernardo Sepúlveda, Centro Médico Nacional Siglo XXI, Instituto Mexicano del Seguro Social, Mexico City 06720, Mexico

**Keywords:** sex steroid hormones, spinal cord injury, estradiol, progesterone, testosterone, neuroprotection

## Abstract

The primary mechanism of traumatic spinal cord injury (SCI) comprises the initial mechanical trauma due to the transmission of energy to the spinal cord, subsequent deformity, and persistent compression. The secondary mechanism of injury, which involves structures that remained undamaged after the initial trauma, triggers alterations in microvascular perfusion, the liberation of free radicals and neurotransmitters, lipid peroxidation, alteration in ionic concentrations, and the consequent cell death by necrosis and apoptosis. Research in the treatment of SCI has sought to develop early therapeutic interventions that mitigate the effects of these pathophysiological mechanisms. Clinical and experimental evidence has demonstrated the therapeutic benefits of sex-steroid hormone administration after traumatic brain injury and SCI. The administration of estradiol, progesterone, and testosterone has been associated with neuroprotective effects, better neurological recovery, and decreased mortality after SCI. This review evaluated evidence supporting hormone-related neuroprotection over SCI and the possible underlying mechanisms in animal models. As neuroprotection has been associated with signaling pathways, the effects of these hormones are observed on astrocytes and microglia, modulating the inflammatory response, cerebral blood flow, and metabolism, mediating glutamate excitotoxicity, and their antioxidant effects. Based on the current evidence, it is essential to analyze the benefit of sex steroid hormone therapy in the clinical management of patients with SCI.

## 1. Introduction

Spinal cord injury (SCI) causes loss of motor, sensory, and autonomic functions and produces severe functional alterations (urinary tract and kidney infections, bowel problems, and cardiac and respiratory dysfunctions). Depending on the severity of the injury, complications often result in the death of SCI patients [1,2,3,4].

To date, consistently effective clinical treatments for SCI are not widely available. Most surgical procedures target the stabilization and decompression of the spinal cord in combination with high doses of methylprednisolone [5]. However, the effects of surgery are limited. Moreover, there is no consensus on the real benefit of methylprednisolone administration, as significant side effects are frequently observed [6]. Therefore, developing reliable repair strategies and treatments for SCI is essential.

The beneficial modulation of synaptic plasticity in spared but reactive neural tissue will significantly differ from the requirements to bridge new axon growth across non-neural lesion cores of anatomically complete lesions. Consequently, different cellular and molecular targets are under investigation to improve the outcome after SCI by providing tissue protection, modulating circuit reorganization, and regulating neural bridging connectivity through the injury site [7].

Increasing experimental and clinical evidence has demonstrated the therapeutic benefits of sex steroid hormones after traumatic brain injury and SCI [8,9,10,11,12,13]. The administration of estradiol (E2) and progesterone (P4) has been associated with a decrease in mortality, better neurological outcomes, and neuroprotective effects after such injuries [10,14,15,16,17,18]. This review aims to analyze the evidence in studies that support hormone-related neuroprotection in SCI and the possible underlying mechanisms in animal models.

## 2. SCI Pathophysiology

SCI undergoes primary and secondary mechanisms of injury [19,20]. The primary mechanisms of injury result from physical forces, including compression, shearing, laceration, and acute stretch/distraction of the spinal cord in the initial traumatic event [21]. A cascade of secondary mechanisms of injury is then initiated, which expands the injured nerve tissue area and exacerbates neurological deficits accordingly [22,23]. Secondary mechanisms in SCI generate delayed and progressive tissue injuries due to the initial injury. Secondary damage can be subdivided into the immediate phase (approximately 2 h post-injury), acute phase (48 h post-injury), subacute phase (2–13 days post-injury), intermediate phase (2–6 weeks post-injury), and chronic phase (>6 months post-injury). Once the chronic phase is established, neurological abnormalities may occur due to axonal damage in orthograde and retrograde directions days to years after SCI [24,25].

During the secondary phase of injury, vascular changes such as hemorrhage, vasospasms, thrombosis, loss of autoregulation, breakdown of the blood-brain barrier, and inflammatory cell infiltrate at the injury site may be observed. Inflammatory cells trigger the release of proinflammatory cytokines, including tumor necrosis factor-α (TNF-α), interleukin (IL)-1α, IL-1β, and IL-6, which reach their maximum levels 6 to 12 h after SCI and remain elevated up to 4 days after injury [26]. In addition, the loss of ionic homeostasis after SCI generates intracellular hypercalcemia, activating calcium-dependent proteases, producing mitochondrial dysfunction, and leading to edema, ischemia, and cell death by apoptosis [27,28,29].

Furthermore, phagocytic inflammatory cells release reactive oxygen species (ROS). Therefore, free radicals react with the polyunsaturated fatty acids of the cell membrane, leading to peroxidation and disruption of the normal phospholipid architecture of cellular and subcellular membranes. Moreover, lipid peroxidation generates aldehyde products that impair the function of critical metabolic enzymes, such as Na^+^,K^+^-ATPase [30]. This enzyme’s activity is essential for maintaining neuronal excitability, so its failure leads to the loss of neuronal function and, ultimately, tissue disruption [31].

Following SCI, an increase in excitatory amino acid release (glutamate and aspartate) from disrupted cells is detected [32,33]. The excessive activation of excitatory amino acid receptors produces excitotoxicity and further loss of neurons and glia by necrotic and apoptotic cell death [34]. Oligodendrocytes are particularly susceptible to loss via apoptotic death at the injury site and in distant regions, leading to the demyelination of the preserved axons [35,36]. The death of oligodendrocytes in white matter tracts continues for several weeks after the injury and may contribute to post-injury demyelination [35]. One of the cellular mediators of apoptosis after SCI is the relationship between microglia and dying oligodendrocytes, suggesting the involvement of microglial activation [37].

The last phase of secondary injury, the chronic phase, comprises events such as white matter demyelination and gray matter dissolution, connective tissue deposition, and reactive gliosis, leading to the formation of a glial scar. The glial scar, composed predominantly of reactive astrocytes, microglia/macrophages, and extracellular matrix molecules, mainly chondroitin sulfate proteoglycans—molecules capable of inhibiting axonal growth—prevents axonal growth by acting as a physical barrier [38,39,40].

## 3. Sex Steroid Hormones

Under physiological conditions, sex steroid hormones (SSHs) mediate various functions such as reproduction, sexual behavior, bone mineralization, and respiration. In the brain, SSHs are involved in neuronal plasticity, memory, and learning processes [41]. Furthermore, SSHs have shown a neuroprotective effect in different models of neuronal damage, including traumatic brain injury (TBI), SCI, ischemic stroke, excitotoxic damage of hippocampal neurons, and neurodegenerative diseases [42,43,44,45].

Some studies have shown that treatments with E2, P4, and testosterone—synthesized from cholesterol—reduce the damage caused by SCI (Figure 1). In animal models, researchers have found gender differences in functional outcomes of SCI, with a markedly better locomotor recovery in female rodents [46,47].

SSHs reduce excitotoxicity, free radical formation, edema, and apoptosis, inhibit inflammatory cytokines, and induce increased angiogenesis, mitochondrial recoupling, and remyelination [48,49] (Figure 2). Treatment with SSHs produces better results in SCI and has limited or more easily anticipated side effects than other treatments. In addition, SSH treatments are relatively accessible and inexpensive, which enhances their potential for widespread use [50].

Several signaling pathways that regulate SSHs are essential for functional recovery. SSH effects are carried out through at least two signaling pathways (previously classified as “genomic” and “non-genomic”). A more appropriate terminology suggested by Hammes and Levin is “nucleus-initiated signaling” for genomic signaling pathways and “membrane-initiated signaling” for fast and non-genomic pathways [51].

On the one hand, one mechanism of action of E2 and P4 involves nucleus-initiated signaling, which is conducted by interacting with specific receptors (ERs α/β and PRs A/B). On the other hand, membrane-initiated signaling is conducted through membrane receptors activating signaling pathways via second messenger cascades, such as those dependent upon nitric oxide (NO) [52,53,54,55]. The rate of protein biosynthesis limits SSH nucleus-initiated signaling and requires more time (minutes to hours) than membrane-initiated signaling modulation, which is faster (milliseconds to seconds) [56,57,58]. Thus, the mechanism of action of SSHs should be considered when timing treatment and counteracting the pathophysiological events that occur in SCI.

Different receptors for SSHs are found in the central and peripheral nervous systems. The classical estrogen receptors (ERs α and β) [59,60], progesterone receptors (PRs A and B) [61,62], and androgen receptors [63] are highly expressed in these systems.

In the spinal cord, ERα expression is higher in ependymal cells and neurons of the dorsal root ganglia and the dorsal horns [64]. Moreover, ERβ expression is lower than ERα in the outer dorsal horn neurons but higher in the deeper layers of the spinal cord [65]. PRs are expressed in motoneurons, glial cells, and ependymal cells of the central canal of the spinal cord [66]. Conversely, progesterone receptor membrane component 1 (PGRMC1) is expressed in neurons of the dorsal horn and ependymal cells lining the central canal [67]. The observation that P4 up-regulates PGRMC1 mRNA and protein levels in the dorsal horn of the injured spinal cord supports the role of PGRMC1 in mediating the protective effects of P4 in this pathology [68]. Androgen receptors are expressed in the motor neurons of the spinal cord [69].

### 3.1. Neuroprotective Effects of Estradiol on Spinal Cord Injury in Animal Models

Although the effect of SSHs as an alternative therapy in SCI patients has not been investigated in depth, several studies in animal models have provided information on the role of these hormones as a treatment for this injury.

Without treatment, the pathophysiological process in acute SCI progresses to chronic neurodegeneration of the spinal cord due to damage caused by the activation of cysteine proteases, such as calpain [70]. Several experiments in animal models of SCI have been conducted with E2 or 17β estradiol (17β-E2). These studies reported that E2 treatment decreases inflammation, reduces apoptosis, improves locomotor function, and decreases neuronal death after SCI in animal models [4,8,9,10,11,14,15] (Table 1).

Methylprednisolone treatment has been widely studied in SCI but has shown adverse side effects and limited efficacy [71,72]. In contrast, low doses of 17β-E2 (5–10 µg) showed no significant side effects or toxicity in rats with acute SCI treated 48 h after the injury. Moreover, E2 reduced pro-inflammatory and proteolytic activities and protected neurons from the penumbra zone in the caudal region of the spinal cord [4].

In addition to reducing some of the parameters produced after SCI, E2 showed a significant improvement in the treated group compared to the control group. Sribnick et al. (2010) administered E2 (4 mg/kg) intravenously 15 min and 24 h after the injury and continued with five daily intraperitoneal doses of E2 (2 mg/kg). These authors reported that moderately severe lesions (40 g*cm) were reduced. The Basso, Beattie, and Bresnahan (BBB) scale evaluation showed significant improvement in E2-treated animals compared to vehicle-treated animals. This difference was observed as early as 3 days after the injury. After 42 days, the mean final scores for E2-treated rats were four points higher (score 13) than vehicle-treated rats (score 9). These scores indicated that E2-treated rats, on average, supported their body weight, performed hindlimb stepping, and coordinated hindlimb/forelimb stepping more often than vehicle-treated rats, which were able, on average, to use the plantar surface of the hindlimb to support weight, but not to perform normal plantar stepping [14].

The immediacy of treatment administration after SCI is essential for the protective effect of E2. Letaif et al. administered 17β-E2 (4 mg/kg) to rats immediately after moderate SCI at T10 (10 g impact rod from a standardized height of 12.5 mm). These authors observed functional motor recovery and neuroprotection in the 17β-E2-treated group from the fourth week after the injury [11].

Other studies demonstrated that administration of E2 (4 mg/kg) in one-day (15 min and 24 h post-injury) and six-day (15 min and 24 h post-injury plus a 2 mg/kg dose for five days) schemes reduced the percentage of spinal cord damage and edema (water filtration) induced in T12 by a 40 g*cm force injury. E2 significantly reduced inflammation as assessed by infiltration of activated (ED2+ and OX-42+ cells) and non-activated (OX-42+ cells) microglia, prevented astroglia reactivity in the gray and white matter of the spinal cord, reduced cyclooxygenase 2 (COX-2) activity, and attenuated neuronal death (probably by inhibition of calpain and caspase-3 activity). In addition, E2 decreased axonal damage and myelin breakdown in the injury region compared to vehicle-treated rats [8,14]. Unfortunately, these authors did not evaluate functional recovery or how E2 regulates various pathophysiological processes in SCI. Therefore, further research is needed to determine whether the tested E2 treatment scheme positively affects functional recovery and not only pathophysiologic events.

Regarding the E2 mechanism, no significant changes were observed in the concentration and expression of the nuclear factor κB (NFκB) and its inhibitor (IκBα), neither in the cytosolic nor the pellet fraction of the lesion segment when analyzing the one-day scheme between the E2- and vehicle-treated groups. However, statistical differences were identified in NFκB and IκBα concentration and expression in the caudal penumbra region between both groups [8]. Conversely, the E2 six-day scheme decreased NF-κB translocation by diminishing its cytosolic decline. Thus, NF-κB levels were not increased in this region, as observed in vehicle-treated animals. Changes in NF-κB were correlated with the reduction of cytosolic IκB-α in SCI animals treated with E2 [14]. Therefore, the mechanism of E2 also depends on the site of the lesion (the caudal or cephalic region).

Another protective mechanism of E2 involves interacting with its receptor to prevent blood-brain barrier breakdown (BSCB). Lee et al. (2015) reported that E2 (300 g/kg) intravenous administration immediately after SCI and 6 and 24 h after the injury at the T9 level prevented BSCB through the inhibition of matrix metalloprotease-9 and reduction of sulfonylurea receptor 1/melastatin transient receptor potential 4 (SUR1/TrpM4) expression [73].

Endogenous hormones could modify the effects of hormonal treatments. The protective effects of E2 on SCI have been reported in intact and gonadectomized male rats. To eliminate endogenous testis-derived androgens, Kachadroka et al. conducted a bilateral gonadectomy in adult Sprague-Dawley rats one week before mid-thoracic crush injury (with forceps). After 30 min post-SCI, animals received different doses of 17β-E2 by implanting a subcutaneous pellet designed to release 0.05, 0.5, or 5.0 mg of 17β-E2 for 21 days. In animals treated with 0.5 or 5.0 mg/kg 17β-E2, they observed improved hindlimb locomotion associated with reduced apoptosis and increased neuronal survival produced and enhanced by the Bax/Bcl-xL protein ratio. Notably, the absence of androgens produced more severe damage than that observed in intact animals, but 17β-E2 treatment reduced SCI-induced apoptosis in gonadectomized and intact rats. These data suggest that E2 may reduce secondary damage after SCI in males [9].

The effects of E2 are concentration-dependent. The administration of high doses of E2 may have some disadvantages, such as gynecomastia, an increased risk of some types of cancer and infertility in females [74], and the development of other types of cancer in males [75]. Therefore, some authors studied the use of nanoparticles for administering low doses of E2 (2.5–25 µg/kg) focally delivered in Sprague-Dawley adult male rats with moderate to severe SCI. E2 nanoparticles diminished the levels of some inflammation factors in plasma, cerebrospinal fluid, and spinal cord tissue, such as leptin, TNF-α, macrophage inflammatory protein-1alpha (MIP-1α), IL-6, IL-4, IL-2, IL-10, interferón γ (IFNγ), and other interleukins [76].

Samantaray et al. performed two studies to evaluate the effect of low doses of acute E2 treatment in a severe injury model of SCI at T10. In the first study, animals received E2 injections (10 or 100 µg/kg) 15 min and 24 h post-injury and were sacrificed 48 h later. Both doses of E2 reduced reactive gliosis, calpain activity, caspase-3 activity, and the Bax/Bcl2 ratio, consequently reducing neuronal death produced by SCI in the penumbra zone in the caudal region and at the lesion site. In the second study, animals were treated for seven days with the same doses of E2 following injury. Protective effects were reported 42 days post-injury: tissue integrity preservation, edema decrease, and inflammation control, significantly reducing glial reactivity, axonal degeneration, and increasing angiogenic markers compared to vehicle-treated rats. Notably, both doses of E2 were equally effective in alleviating the acute damage in these studies [4,77]. As the E2 dose of 10 μg/kg is not physiological, it should be noted that it may cause significant side effects.

Another effort to achieve the beneficial effects of E2 without its side effects is treatment with estrogen receptor modulators such as tamoxifen. Tian et al. (2009) conducted a study with tamoxifen in a SCI model for the first time. They found that the administration of tamoxifen (5 mg/kg) 30 min post-injury improved locomotor activity, attenuated edema and myelin loss by decreasing the production of myelin-associated axonal growth inhibitors, and diminished the number of apoptotic neurons and IL-1β levels by microglial activation when compared with vehicle-treated rats [78].

Furthermore, Mosquera et al. observed that chronic treatment with E2 and tamoxifen improved locomotor recovery. Seven days after ovariectomy, Sprague-Dawley adult rats underwent a moderate spinal cord contusion. Pellets with E2 (3 mg) and tamoxifen (15 mg) implanted for 28 days post-injury improved locomotor recovery as assessed by the BBB test, where the locomotion score of the control group remained below that of the experimental group (10–11 vs. 14). When evaluating the antioxidant effect of E2 and tamoxifen 2 days post-injury, the ROS concentration in the lesioned epicenter of the animals treated with E2 was lower than in the control group. Similarly, tamoxifen reduced oxidative stress in the rostral and epicenter segments of the injured spinal cord evaluated 28 days post-injury. The long-term effect of tamoxifen on locomotor recovery, tissue preservation, and ROS formation suggests that tamoxifen administration is effective for the chronic stages of SCI and could be used as a long-term alternative treatment [16].

Recently, Sánchez-Torres et al. observed that tibolone, a selective tissue estrogen activity regulator (STEAR), significantly increased the amount of preserved tissue and improved the recovery of motor function [79]. Thus, estrogens and compounds with estrogenic properties could be a therapeutic alternative for recovering motor function after SCI.

**Table 1 biomedicines-12-01478-t001:** Neuroprotective effects of estradiol on spinal cord injury in animal models.

SCI Animal Model	Treatment	Evaluated Parameters	Outcome vs. Controls	Conclusions	Author (Year) [Ref]
Male rats with severe SCI at T12	17β-estradiol i.v. injection (4.0 mg/kg weight) 15 min and 24 h post-injury	Inflammation (tissue edema, infiltration of macrophages/microglia and NFkB levels, and myelin integrity)	Reduced edema Decreased inflammation and myelin loss	Estrogen’s multi-active nature, acting as an anti-inflammatory, antiapoptotic, and antioxidant, suggests its potential as a therapeutic agent	Sribnick et al. (2005) [8]
C57/BL/6 mice (males and females) with moderate SCI at T10	No treatment	Injury severity and locomotor function	Significantly higher locomotor scores in females	Gender considerably influences the initial injury severity and the ultimate recovery of motor function after SCI. Recovery is remarkably better in females	Farooque et al. (2006) [47]
Male rats with severe SCI at T12	Tamoxifen i.p. injection (5.0 mg/kg weight) 30 min post-injury	BSCB permeability, tissue edema formation, microglial activation, neuronal cell death, myelin loss, and locomotor testing	Decreased inflammation, apoptosis, and myelin loss Improved functional outcome	Tamoxifen provides neuroprotective effects for SCI-related pathology and disability, making it a potential neuroprotectant for human SCI therapy	Tian et al. (2009) [78]
Male rats with moderately severe SCI at T10	Estrogen i.v. injection (4 mg/kg weight) 15 min and 24 h post-injury, followed by a daily dose (2 mg/kg weight) for 5 days	Inflammation, glial reactivity, neuron death, myelin loss, and locomotor function	Reduced inflammation, neuronal death, myelin loss Prevented glial reactivity Improved locomotor function	Estrogen may help prevent damage and improve locomotor function in chronic SCI	Sribnick et al. (2010) [14]
Male rats with midthoracic crush SCI injury	17β-estradiol s.c. pellet-release (0.05, 0.5, or 5.0 mg) over 21 days	Cell death, expression of Bcl-family proteins, white-matter sparing, and hindlimb locomotion	Increased neuronal survival, white-matter sparing, and hindlimb locomotion Reduced apoptosis	17β-estradiol is an effective therapeutic intervention for reducing secondary damage after SCI in males	Kachadroka et al. (2010) [9]
Male rats with acute SCI at T10	17β-estradiol i.v. injection (1–10 μg/kg) 15 min–4 h post-SCI	Microgliosis and neuronal death	Reduced microglial activation Attenuated apoptosis	Low or physiologic doses of 17-β estradiol reverse secondary pathophysiology in an animal model of SCI through anti-inflammatory and anti-apoptotic actions	Samantaray et al. (2011) [10]
Male rats with acute SCI at T9	17β-estradiol i.v. injection (100 μg/kg weight) 15 min and 24 h after SCI	Neuronal death and functional recovery	Prevented apoptotic cell death Enhanced functional recovery through GPER1	GPER1 may mediate estrogenic neuroprotection in SCI	Hu et al. (2012) [15]
Female rats with moderate contusion SCI at T9–T10	Estradiol (3 mg) or estradiol + tamoxifen (15 mg) silastic implants	Locomotor functional recovery, lesion area, estrogen receptor alpha (ER-α) expression	Estradiol improved locomotor function and up-regulation of ER-α Estradiol and tamoxifen reduced the lesion cavity extension	Estradiol improves functional outcomes mediated by ER-α dependent and independent mechanisms	Mosquera et al. (2014) [16]
Male rats with moderate SCI at T10	17β-estradiol i.p. injection (4 mg/kg weight) immediately after SCI	Functional recovery and motor-evoked potential	Improved functional recovery and motor-evoked potential	17β-estradiol improved neurological and functional motor recovery in rats with SCI	Letaif et al. (2015) [11]
Male rats with moderate contusion SCI at T9	Estradiol i.v. injection (300 g/kg weight) immediately after SCI	BSCB disruption, progressive bleeding, and inflammation	Attenuated BSCB permeability and hemorrhage Reduced neutrophil and macrophage infiltration	Estradiol’s neuroprotective effect after SCI is partially mediated by inhibiting BSCB disruption and hemorrhage	Lee et al. (2015) [73]
Male rats with moderate to severe SCI at T9 and T10	Estradiol nanoparticles (25 µg or 2.5 µg) placed directly on the dural surface of the SCI	Inflammation	Rapid anti-inflammatory effect	Nanoparticle-delivered estrogen may provide a safe and effective treatment option for patients with acute SCI	Cox et al. (2015) [76]
Male rats with moderately severe SCI at T10	17β-estradiol i.v. injection (10 μg/kg weight) 15 min and 24 h post-SCI	Inflammation, neural death, reactive gliosis	Reduced inflammation, neural death, and reactive gliosis	Acute treatment (48-h) with low doses (5–10 μg) of 17β-estradiol attenuates several destructive pathways and brings neuroprotection	Samantaray et al. (2016) [4]
Male rats with moderately severe SCI at T10	17β-estradiol i.v. injection (10 or 100 μg) 7 days post-SCI	Inflammation, cells and axons, and improved locomotor function	Reduced glial reactivity and axonal and myelin damage Increased angiogenic factors expression and microvessel growth Improved locomotor function	Very low doses of 17β-estradiol show significant therapeutic implications for improving locomotor function in chronic SCI	Samantaray et al. (2016) [77]

Bax, protein X associated with Bcl-2; Bcl, B-cell CLL/lymphoma; Bcl-xL, B-cell CLL/lymphoma-extra-large; BSCB, blood-spinal cord barrier; ER, estrogen receptor; ERα, estrogen receptor alpha; GERP1, membrane-bound G-protein coupled estrogen receptor 1; i.p., intraperitoneal; i.v., intravenous; NF κB, nuclear factor κB; s.c., subcutaneous; SCI, spinal cord injury; T, thoracic vertebrae.

### 3.2. Neuroprotective Effects of Progesterone on Spinal Cord Injury in Animal Models

Progesterone can modify some biochemical and molecular processes triggered after SCI or TBI due to its neuroprotective effects [80]. In a SCI transection model, Sprague-Dawley males received daily subcutaneous injections of P4 (16 mg/kg/day for 3 or 21 days). P4 treatment inhibited astrocyte and microglia/macrophage proliferation and activation but stimulated oligodendrocyte precursor cell proliferation and differentiation after SCI. Under these conditions, P4 preserved motor neuron structure, reduced the proliferation and activation of astrocytes and microglia, and increased the production of oligodendrocyte progenitor cells, brain-derived neurotrophic factor (BDNF), and acetylcholine levels, among other processes necessary to restore motor function [81,82,83] (Table 2).

Remyelination is essential for functional recovery after SCI. Several studies have reported P4-promoted remyelination in rats with SCI [18,82,83]. In Schwann cells—the myelinating glia of the peripheral nervous system—P4 activates genes encoding the myelin proteins P0 and peripheral myelin protein 22 (PMP22), induces the expression of Krox 20 (a transcription factor related to myelinogenesis), and enhances the initiation and rate of myelin formation [84]. In the spinal cord, P4 differentiates oligodendrocyte precursor cells (OPC), which proliferate into mature oligodendrocytes 3 days after the lesion and produce myelin proteins such as the myelin basic protein (MBP) and proteolipid protein (PLP) [13,82]. Daily administration of P4 (16 mg/kg) also increases OPC survival, preventing apoptosis by a PR-dependent mechanism 48 h after SCI by transection at T10 [18].

Acting as a differentiation factor, P4 enhanced the expression of the oligodendrocyte lineage transcription factor 2 (Olig2), NK2 homeobox 2 (Nkx2.2), Sry-box 10 (Sox10), and achaete-scute complex homolog-1 (ASCL1, also known as Mash1), determined 3 days after the spinal cord transection at the T9 level [82,85]. Olig2 is a transcription factor involved in oligodendrocyte linage specification and differentiation, which is up-regulated during the differentiation program [86]. OPC differentiation requires up-regulation and co-expression of Olig2 and Nkx2.2 [87]. In turn, Mash 1 induces the expression of Nkx2.2 and, along with Olig2, promotes OPC differentiation in the spinal cord [88]. Simultaneously, Sox10 stimulates the transformation of premyelinating oligodendrocytes into myelinating cells by inducing the expression of MBP and the myelin regulatory factor (MRF), which participates in internodes and myelin formation [89,90].

In addition to increasing the expression of several myelination-related molecules, P4 increased the number of total mature oligodendrocytes, myelin basic protein immunoreactivity, and axonal profiles at the epicenter of the lesion. In 2014, male Wistar rats with moderate-severe contusion SCI at the T8 level received daily subcutaneous injections of P4 (16 mg/kg/day) for 60 days until sacrifice. P4 reduced the volume and rostrocaudal extent of the lesion 60 days after SCI. P4 treatment also significantly improved motor function recovery as assessed by the BBB functional scale and gait recovery as assessed by the Cat-Walk analysis [91].

In 2011, another study highlighted the effect of P4 on myelination. In a model of spinal cord demyelination by intraspinal injection of lysophosphatidylcholine, male mice received a P4 implant (100 mg), which increased circulating levels of steroids comparable to those observed during pregnancy. Seven days later, mice received a single dose of 1% lysophosphatidylcholine in the dorsal funiculus of the spinal cord. One week later, P4 treatment reduced the area of demyelination after injury and inhibited microglia/macrophage activation [92].

In addition to promoting myelination, P4 may exert other neuroprotective effects. The P4 treatment protected cultured spinal cord neurons against glutamate toxicity. In rats with moderately severe SCI, intraperitoneal treatment with P4 (4 mg/kg) 30 min after injury showed a better clinical and histological outcome than controls [93,94]. Furthermore, in rats with complete spinal cord transection at T10, treatment with P4 (4 mg/kg) at one hour (intraperitoneal) and 24, 48, and 72 h (subcutaneous) post-injury showed a significant neuroprotective effect [95].

P4 may exert its neuroprotective effects through several mechanisms. For example, P4 administration also restored the reduced levels of the sodium pump mRNA and choline acetyltransferase (ChAT), whereas the growth-associated protein (GAP-43) mRNA levels were further enhanced [95]. These results are significant because ChAT catalyzes acetylcholine synthesis—essential in muscle contraction, Na^+^,K^+^-ATPase maintains the membrane potential, neuronal excitability, and entry of metabolites and ions into the soma, whereas GAP-43 is involved in axonal regeneration [12,17,81]. The expression of BDNF mRNA and protein levels increased in ventral horn motoneurons, where P4 prevented the lesion-induced chromatolysis degeneration of spinal cord motoneurons as determined by Nissl staining [17]. P4 also enhanced the tropomyosin receptor kinase B (TrkB) neurotrophin receptor and phosphorylated cAMP-responsive element-binding protein (pCREB) immunoreactivity in motoneurons [12,96], and partly normalized the ultrastructural abnormalities, and preserved the microtubule-associated protein 2 (MAP2) immunostaining of deafferented motoneurons. As P4 treatment up-regulates MAP2 staining in dendrites and perikaryon, it is suggested to act on the cytoskeleton [97].

The neuroprotective effect of P4 has also been demonstrated with organotypic cultures of spinal cord slices from 3-week-old mice. Using a weight drop model, a decrease in the number of motoneurons was induced by in vitro SCI, which correlated with an increase in the number of dying cells and lactate dehydrogenase (LDH) release. When 10 µM of P4, 5α- dihydro-progesterone (5α-DHP), or allopregnanolone (3α, 5α-tetrahydro-progesterone) were added at the time of injury, the spinal cord slices were rescued from the effects of damage. These authors demonstrated that the neuroprotective effects of P4 are PR-dependent, as these neuroprotective effects were not observed in slices of homozygous knockout PR−/−mice [98].

Interestingly, Yang et al. (2017) conducted a study evaluating the mechanism by which P4 significantly reduced axonal dieback and neuronal death in mice following SCI. These authors demonstrated that this effect of P4 was mediated through the down-regulation of pro-inflammatory cytokines, including inducible nitric oxide synthase (iNOS), monocyte chemoattractant protein-1 (MCP-1), and IL-1β, and activation of caspase-3 and glial fibrillary acidic protein (GFAP). Repeated imaging showed that axonal dieback distance was significantly reduced in mice treated with P4 (16 mg/kg) after SCI by hemisection at the T11 level. The densities of astrocytes and microglia were significantly higher in the vehicle-treated group than in the P4-treated group. Moreover, P4 also improved behavioral performance post-injury. These findings suggest that P4 exerts a neuroprotective effect by attenuating axonal dieback, reducing the activation and proliferation of astrocytes and microglia, and inhibiting the release of pro-inflammatory cytokines [99].

Regarding neuroinflammation, P4 decreases the activation and proliferation of astrocytes and microglial cells in the acute and chronic phases of SCI. Moreover, P4 therapy regulates astrocytes and microglial cells by suppressing neuroinflammation and creating a pro-differentiating environment, which results in neurological recovery after SCI. Thus, P4 is a glioactive factor that favors remyelination and inhibits reactive gliosis [18,83].

In 2015, Labombarda et al. showed that P4 down-regulated pro-inflammatory cytokines, such as IL1β1, IL6, and TNFα, and enzymes involved in ROS production, such as COX-2 and iNOS, through a PR-dependent mechanism via NFkB inhibition [18]. Sprague-Dawley male rats received daily subcutaneous injections of P4 (16 mg/kg) after SCI by transection at the T10 level and were sacrificed 6 h, 24 h, or 48 h, and 3 or 21 days after surgery. P4 significantly attenuated the injury-induced hyperexpression of IL1β1, IL6, TNFα, iNOS, and COX-2 mRNAs, all involved in oligodendrocyte damage. Consequently, P4 down-regulated reactive gliosis and exerted potent anti-inflammatory effects in rats with SCI [18].

Furthermore, SCI is frequently associated with the development of chronic pain. Therefore, different studies have investigated the effects of P4 on the development of allodynia after SCI. In Sprague-Dawley male rats (200–220 g), the spinal cord was unilaterally hemisected at the T13 level, and natural P4 was administered (16 mg/kg/day) immediately after the lesion and once a day until the animals were euthanized (either at 1, 14, or 28 days post-injury). P4 administration prevents neuropathic pain-associated behaviors [100,101,102], modulates the expression of NMDAR subunits and the upregulation of protein kinase C γ (PKC γ) [100], and attenuates the proinflammatory cascade induced by SCI, which involves pro-inflammatory enzymes such as COX-2 and iNOS [101], and different cytokines [102], all key players in neuropathic pain generation, probably by reducing NFĸB transactivation potential [101]. In addition, early and sustained P4 treatment prevents mechanical and thermal allodynia in animals subjected to SCI [103].

Tissue preservation in the spinal cord may be related to functional recovery. After a moderate-intensity SCI in mature Sprague-Dawley male rats, the intraperitoneal administration of P4 (4 mg/kg) demonstrated neuroprotective effects over 5 days post-injury. The treatment was initiated 30 min after injury and repeated at 6, 24, 48, 72, 96, and 120 h intervals. Thomas et al. observed a significant reduction in the central cavitary process that follows SCI, accompanied by locomotor score improvement [94]. Therefore, P4 administered a few minutes after SCI promotes spinal cord tissue preservation and motor function recovery.

Interestingly, some studies have reported that the beneficial effects of P4 on SCI are inconclusive. For example, based on functional and histological analyses, Calvacante et al. observed that P4 acute administration (4 mg/kg) 30 min before transient occlusion of the proximal descending thoracic aorta of male rats failed to prevent or attenuate ischemia SCI, as no significant differences were found among the studied groups (17β-E2, P4, or vehicle). An initial significant impairment of hind limb motor function was observed in all the groups, with partial improvement over time but no significant differences among groups during the observation period. Similarly, gray matter analysis showed a lack of viable neurons, and the number of viable neurons per section showed no differences among these groups [104].

Furthermore, Fee et al. found no improvement with P4 injection following an SCI by contusion. Both the short-term (5 days of either 4 or 8 mg/kg) and long-term (14 days of either 8 or 16 mg/kg) therapies failed to show any significant alteration in locomotor functioning and injury morphometrics after 21 days [105]. Part of this discrepancy could be the duration of the treatment; i.e., treatments in the study of Fee et al. [105] were limited to 5 and 14 days, while treatments in studies with positive results lasted for up to 60 days [91,102,103].

**Table 2 biomedicines-12-01478-t002:** Neuroprotective effects of progesterone on spinal cord injury in animal models.

SCI Animal Model	Treatment	Evaluated Parameters	Outcome vs. Controls	Conclusions	Author (Year) [Ref]
Male rats with moderate SCI	P4 i.p. injection (4 mg/kg weight) 30 min after SCI and repeated at 6 h, 24 h, 48 h, 72 h, 96 h, and 120 h intervals	Injury severity and locomotor function	Improved BBB scores Tissue preservation	P4 showed potential therapeutic properties in managing acute SCI	Thomas et al. (1999) [94]
Male rats with complete spinal cord transection at T10	P4 i.p. injection (4 mg/kg) 1 h after SCI, and s.c. administration 24, 48, and 72 h post-injury	Neuronal function under negative regulation (ChAT and Na,K-ATPase) and stimulated neuronal function (GAP-43)	Restored ChAT immunoreactivity and a3 catalytic and b1 regulatory subunits of neuronal Na/K-ATPase mRNA Enhanced GAP-43 mRNA in ventral horn neurons	P4 appears to replenish acetylcholine, restore membrane potential, ion transport, and nutrient uptake, and accelerate reparative responses to injury	Labombarda et al. (2002) [95]
Male rats with complete spinal cord transection at T10	P4 i.p. injection (4 mg/kg) 1 h after SCI, and s.c. administration 24, 48, and 72 h post-injury	Expression of PR and 25-Dx binding proteins for P4	PR mRNA levels did not change Increased 25-Dx mRNA	Distinct membrane-binding sites may mediate P4 neuroprotective effects	Labombarda et al. (2003) [68]
Male rats with complete spinal cord transection at T10	P4 s.c. administration (4 mg/kg) 1 h and again at 24, 48, and 72 h post-injury	Expression of BDNF at both the mRNA and protein levels Chromatolysis analysis	Increased BDNF expression (mRNA and protein levels) in ventral horn motoneurons Prevented the chromatolysis degeneration of spinal cord motoneurons	P4 increased neuronal BDNF, which could provide a trophic environment and might be part of the P4-activated pathways to provide neuroprotection	González et al. (2004) [17]
Male rats with SCI at T10	P4 s.c. administration (4 mg/kg) 1 h and again at 24, 48, and 72 h post-injury	Expression of the BDNF mRNA and BDNF immunoreactivity receptor TrkB Chromatolysis analysis MBP expression at the mRNA and protein levels PR expression	Increased BNDF (mRNA and protein expression) in ventral horn motoneurons Increased TrkB and pCREB in motoneurons Prevented lesion-induced chromatolysis Restored myelination	P4-induced BDNF expression might regulate the function of neurons and glial cells in a paracrine or autocrine fashion and prevent the generation of SCI damage	De Nicola et al. (2006) [12]
Male rats with SCI at T10	P4 s.c. administration (4 mg/kg) 1 h and again at 24, 48 and 72 h post-injury	Expression of MBP at the mRNA and protein levels NG2-immunopositivity as markers for OPCs RIP-immunopositivity as mature oligodendrocytes identifier	Increased MBP expression (mRNA and protein) in the CST and DAT but not VF Stimulated NG2-immunostaining in the gray and white matter. Unchanged RIP-positive cells	P4 effects on MBP expression and NG2 immunopositivity may contribute to neuroprotection	Labombarda et al. (2006) [13]
Male and female rats with moderate spinal cord contusion at T10	Short-term (5 days) P4 (4 or 8 mg/kg) and long-term (14 days) P4 (8 or 16 mg/kg)	Locomotor recovery Morphologic assessment of white and grey matter	No changes in locomotor recovery No differences in spinal cord morphology.	This study does not support P4 therapy as a potential therapeutic agent in SCI	Fee et al. (2007) [105]
Male rats with complete spinal cord transection at T10	P4 s.c. administration (16 mg/kg/day) for 3 or 21 days after injury	OPC parameters (NG2 immunostaining), mature oligodendrocytes, and central myelin proteins (MBP, PLP) Oligodendrocyte transcription factors (Olig1, Olig2, and Nkx2.2) Myelin proteins (MBP and PLP-immunoreactivity)	Short treatment (3 days) Increased the number of OPC Increased MBP expression (mRNA and protein) Increased Olig2 and Nkx2.2 (mRNA) Prolonged treatment (21 days) Enhanced maturation and differentiation of oligodendrocytes Increased PLP expression (mRNA and protein) Increased Olig1 (mRNA)	Short P4 treatment influenced the proliferation and differentiation of OPC into mature oligodendrocytes. Prolonged P4 treatment favored remyelination and oligodendrocyte maturation. Thus, P4 effects on oligodendrogenesis and myelin proteins may constitute fundamental steps for repairing traumatic injuries to the spinal cord	Labombarda et al. (2009) [82]
Male rats unilaterally hemisected at T13	P4 s.c. administration (16 mg/kg/day)	Behavioral evaluation of mechanical and cold allodynia Expression of NMDAR subunits (NR1, NR2A, NR2B), PKCγ, ppD, and KOR	Prevented the development of mechanical allodynia Reduced the painful responses to cold stimulation Prevented NMDAR subunits and PKCγ mRNA upregulation No changes in ppD expression (mRNA) Increased KOR expression	P4 modulates neuropathic pain after SCI, creating a favorable molecular environment that may decrease spinal nociceptive signaling	Coronel et al. (2011) [100]
Primary demyelination model in male mice	Single implant of P4 (100 mg).	Determination of total myelin and MBP Determination of OX-42+ microglia/macrophages Staining of oligodendrocyte precursors (NG2+ cells) and mature oligodendrocytes (CC1+ cells) Analysis of the microglial marker CD11b mRNA	Decreased area of demyelination by 50% Increased oligodendrocyte precursors and mature oligodendrocytes Decreased OX-42+ cells Reduced CD11b (mRNA)	P4 exerts promyelinating and anti-inflammatory effects at the spinal cord level	Garay et al. (2011) [92]
Male rats with complete spinal cord transection at T10	P4 s.c. administration (16 mg/kg/day) for 3 or 21 days after injury	Immunodetection of S100β, GFAP, NG2+ oligodendrocyte precursors, CC1+ oligodendrocytes, and OX-42+ microglia/macrophages	Acute treatment (3 days) Inhibition of astrocyte proliferation and activation Acted as a pro-myelinating factor Chronic treatment (21 days) Inhibited reactive astrogliosis Decreased NG2+ cells EnhancedCC1+ cells	P4 emerges as a glioactive factor, favoring remyelination and inhibiting reactive astro- and microgliosis	Labombarda et al. (2011) [83]
Male rats unilaterally hemisected at T13	P4 s.c. administration (16 mg/kg/day)	Expression and activity of spinal COX-2 and iNOS IκB-α mRNA levels Profile of glial cell activation Pain-associated behaviors	Reduced COX-2 and iNOS (mRNA levels) Decreased COX-2 activity and nitrite levels Lower IκB-α levels (mRNA) Attenuated GFAP and OX-42 positive cells increase Prevented the development of mechanical and thermal allodynia	P4 may represent a valuable strategy to prevent the development of central chronic pain by modulating early neuroinflammatory events after SCI	Coronel et al. (2014) [101]
Male rats with moderately severe SCI at T8	P4 s.c. administration (16 mg/kg/day) for 60 days	Tissue preservation using magnetic resonance imaging and quantification of tissue-sparing Optical density of MBP staining Total number of APC+ cells Quantification of axonal profiles Functional outcome evaluated with the BBB scale Sensory function (mechanical and thermal sensitivity)	Reduced volume and extension of the lesion Increased the number of mature oligodendrocytes MBP, immunoreactivity, and the number of axonal profiles at the epicenter of the lesion Improved locomotor outcome	P4 beneficial actions on locomotor outcome could be related to the reduction of secondary damage and the preservation or regeneration of axons and myelin of the descending pathways	García-Ovejero et al. (2014) [91]
Male rats with complete spinal cord transection at T10	P4 s.c. administration (16 mg/kg/day). Animals were euthanized 6 h, 24 h, 48 h, 3 days, or 21 days following surgery	Expression of proinflammatory factors and enzymes	Attenuation of IL1β, IL6, TNFα, iNOS, and COX (mRNAs)	PR participates in the anti-inflammatory effects of P4, the modulation of astrocyte and microglial responses, and the prevention of OPC apoptosis	Labombarda et al. (2015) [18]
Male PRKO mice (inactivated PRA and PRB isoforms) with complete spinal cord transection at T10		Immunohistochemistry to assess astrocytes, microglia, and OPC Detection of OPC apoptosis	No decrease in the expression of IL1β, IL6, TNFβ, and IkB-α (mRNAs) No changes in OPC density and OPC apoptotic death		
Male rats unilaterally hemisected at T13	P4 s.c. administration (16 mg/kg/day) immediately after SCI and during 1 or 28 days after injury	Behavioral evaluation of mechanical and cold allodynia Expression of IL-1β, IL-1RI and IL-1RII, IL-1ra, IL-6, and TNFα IL-1β protein levels Immunofluorescence to detect NR1 subunit of NMDAR, IL-1β, NeuN	No mechanical allodynia Reduced sensitivity to cold stimulation Reduced expression of proinflammatory cytokines (mRNA) Reduced IL-1RI-positive neurons in the superficial laminae of the spinal cord Decreased NR1+ in dorsal horn cells	By modulating the expression of pro-inflammatory cytokines and neuronal IL-1RI/NR1 colocalization, P4 emerges as a promising agent for preventing chronic pain after SCI	Coronel et al. (2016) [102]
Male rats unilaterally hemisected at T13	P4 s.c. administration (16 mg/kg/day) immediately after SCI and during 1 or 28 days after injury	Behavioral evaluation of mechanical and cold allodynia Expression of galanin, GalR1, GalR2, NPY, Y1R, Y2R CyCB	Early phase (1 day) No upregulation of Y1R and decrease in ↓NPY, Y2R, and GalR1 levels (mRNA) Chronic phase (28 days) Basal levels of galanin, GalR1, NPY, and Y1R (mRNA) Reduced levels of Y2R (mRNA) No mechanical allodynia Reduced sensitivity to cold stimulation	Early and sustained P4 administration prevents temporal changes in the spinal expression of galanin and NPY and their associated receptors, which could potentially prevent and treat chronic pain after central injuries	Coronel et al. (2017) [103]
Male YFP-H and male CX3CR1^GFP/+^ transgenic mice with hemisected spinal cords at T11	P4 i.p. injection (16 mg/kg) one-hour post-injury and s.c. injection at 3 h, 24 h, and 48 h after SCI	Axonal dynamics and survival neurons Identification of neurons, microglia, and astrocytes Protein levels of caspase-3, GFAP, and MBP mRNA expression of IL-1β, iNOS, and MCP-1 Behavioral function	Attenuation of axonal degeneration and neuronal death Reduced proliferation of microglia No accumulation of astrocytes Reduced expression of pro-inflammatory cytokines, GFAP, and caspase-3 Upregulation of MBP expression Improved behavioral performance	P4 exerted a neuroprotective effect by attenuating axonal dieback, reducing the accumulation of astrocytes and microglia, and inhibiting the release of pro-inflammatory cytokines	Yang et al. (2017) [99]
Male rats with transitory occlusion of the proximal descending thoracic aorta	P4 (4 mg/kg) intra-arterial administration	Motor function Neuronal cell death in grey matter Apoptosis (Bcl-2 and annexin V) Necrosis (propidium iodide)	No significant differences in motor function impairment or the number of viable neurons No changes in apoptosis and necrosis proteins	Acute P4 administration could not prevent or attenuate spinal cord ischemic injury based on functional and histological outcomes	Cavalcante et al. (2018) [104]
Male rats with complete spinal cord transection at T9	P4 s.c. administration (16 mg/kg/day) for 3 days. The first injection was given immediately after SCI	Expression of transcriptional inhibitors (Id2, Id4, hes5) and activators (Olig2, Nkx2.2, Sox10, and Mash1) Immunostaining of OPC, astrocytes, and microglial cells, and double labeling of TGFβ1 and Olig2	Increased transcriptional activator levels (mRNA) Higher density of Olig2-expressing OPC Increased TGFβ1 and the number of astrocytes and microglial TGFβ1 expressing cells	P4 differentiating effects might involve TGFβ1, indirectly mediating these actions by releasing microglial and astrocytic TGFβ1	Jure et al. (2019) [85]

APC, adenomatus polyposis coli; BBB, Basso, Beattie, and Bresnahan scale; BDNF, brain-derived neurotrophic factor; BMS, Basso Mouse scale; ChAT, choline acetyltransferase; COX-2, cyclooxygenase 2; CST, corticospinal tract; DAT, dorsal ascending tract; 25-Dx, progesterone binding protein 25-Dx; EC, eriochrome cyanine; GAP-43, growth-associated protein 43; GFAP, glial fibrillary acidic protein; GFP, green fluorescent protein; iNOS, inducible nitric oxide synthase; i.p., intraperitoneal; KOR, kappa opioid receptor; LFB, Luxol Fast Blue; LPC, lysophospatidylcholine; MBP, myelin basic protein; MCP-1, monocyte chemoattractant protein-1; MOG, myelin oligodendrocyte glycoprotein; NMDAR, N-methyl-D-aspartate receptor; OPC, oligodendrocyte-precursor cells; P4, progesterone; pCREB, phosphorylated cAMP-responsive element binding protein; PKCγ, protein kinase C gamma; PLP, proteolipid protein; ppD, preprodynorphin; PR, progesterone receptor; PRKO, PR knockout; RIP, receptor interacting protein; s.c., subcutaneous; SCI, spinal cord injury; TGF-β1, transforming growth factor beta 1; TNFα, tumor necrosis factor alpha; T, thoracic vertebrae; TrkB, tropomyosin receptor kinase B; VF, ventral funiculus; YFP, yellow fluorescent protein.

### 3.3. Neuroprotective Effects of Androgens on Spinal Cord Injury in Animal Models

Inflammation promotes the exacerbation of damage in the pathophysiology of SCI. Therefore, the regulation of the immune system could control inflammation in SCI. Rouleau et al. provided evidence of the correlation between testosterone levels and immune deficiencies in acute SCI. In CD1 mice studied for 4 weeks after an SCI by transection between the 9th and 10th thoracic vertebrae, a decrease in serum testosterone levels was identified during the first 2 weeks, and an increase in growth hormone (GH) levels occurred one week after the injury. These changes were correlated to a decrease in total leukocyte, lymphocyte, and eosinophil counts in the blood and lymphocytes in the bone marrow. Significant differences were also observed in the hormonal and immune systems, providing evidence for the role of hormones (i.e., GH, insulin, parathyroid hormone, and dehydroepiandrosterone) in immune dysfunction following SCI. Thus, this study highlights a direct correlation between reduced serum testosterone levels and immune dysfunction after SCI [106]. Moreover, inflammation is an essential host defense mechanism. However, although its effects are contradictory—inflammation is central in modulating the pathological progression of acute and chronic SCI but also regulates neuronal damage and regeneration—testosterone could influence both processes [107].

Furthermore, the immune response to trauma also shows sexual dimorphism [108]. In this regard, Hauben et al. demonstrated that male mice and rats show worse locomotor recovery after incomplete SCI than their female littermates. Evaluation with the BBB locomotion scale after a severe spinal cord contusion showed that hindlimb motor performance was significantly better in females [46]. Additionally, these authors reported that castrated males recover better than intact males and females treated post-injury with dihydrotestosterone. Notably, post-traumatic administration of flutamide (a testosterone antagonist) produced a better recovery in male rats [46]. These results could be explained by the inhibitory effect of testosterone on T cell-mediated immunity, suggesting that testosterone may contribute to poor recovery in males with SCI.

To learn whether testosterone could regulate any of the pathophysiological events in SCI, Gürer et al. (2015) showed that a single intraperitoneal dose of testosterone (15 mg/kg) decreased the activity of caspase-3, myeloperoxidase, and xanthine oxidase enzymes. Testosterone also decreased malondialdehyde levels, whereas catalase levels increased following transient global SCI by ischemia in rabbits. Testosterone was adminsistered immediately after the occlusion clamp of the aorta. A mixture of four testosterone esters (testosterone propionate, testosterone phenylpropionate, testosterone isocaproate, and testosterone decanoate) with different half-lives was preferred to provide more stable serum testosterone levels [109]. After testosterone administration, caspase-3, myeloperoxidase, and xanthine oxidase enzyme activity decreased, as did malondialdehyde levels, while catalase levels increased. The authors concluded that testosterone exhibits significant neuroprotective activity after a spinal cord ischemia-reperfusion injury.

In addition to its effects on the spinal cord, androgen treatment attenuates sublesional muscle loss and other phenotypic changes associated with impaired muscle function after SCI [7,110,111,112,113,114] and may promote a slight improvement in locomotor recovery in rodent models [115]. Byers et al. (2012) implanted testosterone-filled Silastic capsules in female rats after SCI at the T9 level. Four weeks later, testosterone treatment prevented the decrease in the dendritic length of quadriceps motoneurons observed in untreated SCI animals. Similarly, the vastus lateralis muscle weights and fiber cross-sectional areas of untreated SCI animals were smaller, and testosterone treatment also prevented these reductions. However, testosterone showed no effect on reducing lesion volume or increasing tissue sparing [113]. These findings support the role of testosterone as a neurotherapeutic agent (Table 3).

**Table 3 biomedicines-12-01478-t003:** Neuroprotective effects of androgens (testosterone) on spinal cord injury.

SCI Animal Model	Treatment	Evaluated Parameters	Outcome vs. Controls	Conclusions	Authors (Year) [Ref]
Male and female rats with severe spinal cord contusions at T8	No treatment	Functional recovery (locomotor hindlimb performance)	Better hindlimb motor performance in females than in males	The better spontaneous recovery of female rats and mice after SCI than that of males is related to the suppressive effect of androgens on the ability to sustain a T-cell-mediated protective response to a CNS insult	Hauben et al. (2002) [46]
Male and female rats with severe or mild spinal cord contusions at T8	No treatment	Functional recovery (locomotor hindlimb performance) Neurological function Morphological analysis of the lesion site	Better locomotor hindlimb function (BBB score) and neurological function (inclined plane test) in females than in males More preservation of neural tissue, better tissue organization, and continuity of myelinated fibers in females than in males	The better functional recovery observed in females may be attributable to improved tissue preservation, possibly due to endogenous neuroprotective processes that do not occur in males	
Male and female WT and nude Balb/c mice with SCI at T12	No treatment	Functional recovery	Significantly better functional recovery of WT females than WT males No differences between nude mice Worse functional recovery in nude mice than WT mice in both genders	T-cell immune response helps the body overcome the effects of destructive self-compounds that emerge from injured tissues	
Male nude Balb/c mice with SCI at T12	Castrated	Functional recovery	Significantly better locomotor performance in castrated mice than their sham-operated littermates	Sexual dimorphism observed in functional recovery from ISCI may, at least partially, be androgen-dependent	
Male rats with mild SCI at T8	Castrated	Functional recovery	Significantly better locomotor performance in castrated rats than their sham-operated littermates		
Female rats with severe SCI at T8	DHT (100 mg) s.c pellet (21-day-release) 10 days post-injury	Functional recovery	Detrimental effects on functional recovery	DHT has an adverse impact on SCI recovery	
Male rats with severe SCI at T8	Flutamide (testosterone-antagonist) i.p. injection (25 mg/kg weight) immediately after SCI and 5 mg/kg every other day for ten days	Functional recovery	Significantly better functional recovery	Testosterone has an adverse effect on SCI recovery	
Male mice with spinal cord transection at T9/10	No treatment	Serum levels of testosterone, GH, PTH, DHEA, insulin, and a complete immune cell count from blood and bone marrow samples	Two weeks after SCI: Decreased serum testosterone levels Increased serum GH levels Three to four weeks after SCI: Normal serum testosterone and GH levels Overall: Reduced DHEA, PTH, and insulin levels Decreased number of total leukocytes, lymphocytes, and eosinophils in blood and bone marrow	Significant changes occur rapidly (<1–2 weeks) in both the hormonal and immune systems after SCI	Rouleau et al. (2007) [106]
Young adult female rats with SCI at T9	Testosterone-filled Silastic capsules	Soma volume, motoneuron number, lesion volume, and tissue-sparing	Prevention of decrease in dendritic length of quadriceps motoneurons, vastus lateralis muscle weights, and fiber cross-sectional areas	Regressive changes in motoneuron and muscle morphology were observed with testosterone treatment	Byers et al. (2012) [113]
Male rats with complete spinal cord transection	Low (2.8 mg/kg) or high (7.5 mg/kg) 24 h doses of testosterone provided with an Alzet pump	Expression of MAFbx, MuRF1, REDD1, and FOXO1 Weight of muscles with different fiber-type compositions	High-dose Reduced adverse effects of MP on body weight, muscle weights, and muscle gene expression	High-dose of testosterone partially protected against muscle atrophy and gene expression changes caused by MP	Wu et al. (2012) [112]
Male rabbits with ischemia/reperfusion SCI	Testosterone i.p. injection (15 mg/kg)	Malondialdehyde and catalase levels Activities of caspase-3, myeloperoxidase, and xanthine oxidase Histopathological, ultrastructural, and neurological studies	Decreased activities of caspase-3, myeloperoxidase, malondialdehyde, and xanthine oxidase Increased catalase levels Improved histopathological scores, ultrastructural scores, and Tarlov scores	Biochemical, histopathological, ultrastructural, and neurological examination findings showed that testosterone has neuroprotective effects on ischemia/reperfusion SCI	Gürer et al. (2015) [109]
Young adult female rats with SCI at T9	DHT-filled Silastic capsules	Functional recovery Lesion volume and tissue sparing, quadriceps muscle fiber cross-sectional area, and motoneuron dendritic morphology	Significantly improved voiding behavior Attenuation of motoneuron dendritic length-SCI-induced decrease Prevented muscle fiber cross-sectional area reductions	DHT treatment ameliorated deficits in micturition and regressive changes in motoneuron and muscle morphology seen after SCI	Sengelaub et al. (2018) [49]
Clinical studies					
Men with SCI	No treatment	Serum levels of FSH, LH, testosterone, estradiol, and PRL The LHRH stimulation test Semen analysis and testicular volumes	Low serum gonadotrophin (LH and FSH) Low serum testosterone Hyperprolactinemia Low semen volume, sperm motility, and normal sperm morphology	SCI patients showed hypogonadotropism due to secondary neural or hormonal pathway alteration, leading to semen quality impairment	Naderi and Safarinejad (2003) [116]
Men with SCI	No treatment	Testosterone and LH serum levels Free testosterone levels Level of disability (FIM instrument and ASIA exams)	High serum LH No changes in total or free testosterone levels Lower testosterone levels in the time since injury < 12 months group Higher testosterone levels in the time since injury >12 months group Lower free testosterone levels in the time since injury ≤ 12 months group	A negative androgen status is notable, especially in the first year after a spinal cord injury Testosterone substitution therapy should be considered during the first year after injury to induce neural regeneration and preserve muscle strength	Celik et al. (2007) [117]
Men with SCI	No treatment	Serum testosterone levels Time since SCI Selected laboratory values	Low testosterone levels, low hemoglobin, and high prolactin were associated with less time since injury	Men with SCI are at risk of low serum testosterone	Clark et al. (2008) [118]
Men with SCI	Testosterone-cypionate (200 mg) i.m. injection, monthly	Motor function (ASIA motor index discharge and FIM total discharge scores)	Improved motor scores in ASI (grades C and D) No differences in discharge FIM scores	Muscle size and strength increased with testosterone	Clark et al. (2008) [119]
Men with chronic SCI	No treatment	Serum total testosterone, albumin, LH, FSH, and prolactin levels	Testosterone deficiency was associated with the severity of the injury and narcotic medication for pain No changes in LH, FSH, prolactin, and albumin levels between incomplete and complete SCI patients	Testosterone levels were significantly associated with the severity of SCI	Dunga et al. (2011) [120]
Men with SCI	No treatment	Serum testosterone, insulin, triglyceride levels HOMA-IR, BMI Hypogonadism-related symptoms (AMS questionnaire) LTPA	Low testosterone levels Increased BMI, insulin, HOMA-IR, triglycerides values Less h/week of LTPA Weekly LTPA and BMI were associated with testosterone levels Decreased sexual desire sub-score of the AMS questionnaire	Poor LTPA, high BMI, and low sexual desire are independent predictors of low testosterone levels in men with chronic SCI	Barbonetti et al. (2014) [121]
Men with SCI	No treatment	Testosterone levels by decade of life	Decline in total serum testosterone (0.6% per year) related to age	Low serum total testosterone concentration occurs earlier in life in men with SCI, with a higher prevalence by a decade of life	Bauman et al. (2014) [122]

ASIA, American Spinal Injury Association; BBB, Basso, Beattie, and Bresnahan scale; BMI, body mass index; DHEA, dehydroepiandrosterone; DHT, dihydrotestosterone; F, female; FIM, Functional Independence Measure; FSH: follicle-stimulant hormone; GH, growth hormone; HOMA-IR, homeostatic model assessment of insulin resistance; IM, intramuscular; i.p., intraperitoneal; LH, luteinizing hormone; LHRH, LH-releasing hormone; LTPA, leisure time physical activity; M, male; MP, methylprednisolone; PRL, prolactin; PTH, parathyroid hormone; SCI, spinal cord injury; T, thoracic vertebrae; WT, wild type.

## 4. Neuroprotective Effects of Sexual Hormones on Spinal Cord Injury in Humans

Although research on hormone replacement as a treatment for SCI has been developed, this type of therapy has not been performed in the clinical phase. Only a few studies on the effects of sexual hormones have been conducted in humans with any SCI condition. Among these studies are those using androgen treatment to correlate testosterone levels with SCI.

Some reports indicate an association between SCI clinical characteristics and low testosterone levels. In two clinical studies, a considerable percentage of men with SCI (60% and 43.3%, respectively) showed low serum testosterone levels [118,120]. Durga et al. (2011) reported an inverse association between low serum testosterone and the severity of SCI. In this study, the participants with complete motor injuries presented a more significant testosterone deficiency than those with incomplete motor injuries [120].

Low testosterone levels in men with SCI have been associated with lower hemoglobin and higher prolactin levels in healthy subjects. From a sample of 102 men with SCI, 60% had low testosterone levels. The mean testosterone concentration was 220 ng/dL (normal reference range: 241 to 827 ng/dL). Testosterone levels < 241 ng/dL were considered abnormally low. Less time since injury was also correlated with low testosterone levels; those men with low testosterone levels were more likely to be acutely injured than men with higher testosterone levels [118].

Androgen levels decrease more notably during the first year after SCI. Celik et al. (2007) conducted a prospective case study that included 44 male patients with SCI. Values from the control group of healthy individuals were considered as references. Testosterone and free testosterone concentrations were evaluated in both groups. Total testosterone levels were significantly higher in SCI patients with a time since injury > 12 months than in the control group. Free testosterone levels were lower in SCI patients with time since injury < 12 months than in patients with time since injury >12 months. Consequently, these authors concluded that during the first year after SCI, patients could show a negative androgenic status [117].

Another study found a similar correlation between low testosterone levels and advancing age in men with SCI. Bauman et al. (2014) found lower serum testosterone levels earlier in SCI patients and a higher prevalence per decade compared to the general population [122]. In the same year (2014), Barbonetti et al. found that one-third of patients showed biochemical androgen deficiency defined by a total testosterone concentration < 300 ng/dL. In the search for the determinants of androgen deficiency, these authors found that low leisure-time physical activity, high body mass index (BMI), and low sexual desire are independent predictors of low testosterone in men with chronic SCI [121]. Overall, these results indicate that the severity of injury, time since injury, and leisure-time physical activity are correlated with androgen deficiency.

Some studies associate the increase of systemic and metabolic complications, such as obesity, cardiovascular diseases, diabetes, osteoporosis, and severe infections, with chronic immobilization due to SCI and changes in hormone levels that can regulate metabolic functions, such as testosterone [118,120,123].

In humans, testosterone increases energy, improves muscle mass and strength, enhances cognition, modifies mood/libido, and sexual function, increases bone mineral density, and has anti-inflammatory effects [123,124]. Consequently, low testosterone levels in patients with SCI lead to other complications, such as alterations in mood/libido, sexual function, and fertility problems. Naderi and Safarinejad (2003) reported low serum testosterone in 16% of men with SCI. In addition, they found hypogonadotropism secondary to the alteration in the hypothalamus/pituitary neural/hormonal pathway in these patients. This mechanism could contribute to the altered semen quality reported in men with SCI [116].

Decreased musculoskeletal integrity and neuromuscular impairment are also significant complications of SCI. Men with low testosterone exhibit low muscle mass, impaired muscle function [125], and worsened walking biomechanics [126]. In contrast, testosterone replacement therapy improves muscle mass, neuromuscular function [127], and walking speed in older ambulatory hypogonadal men [128].

During the first year after injury, when the most significant neurological recovery occurs, testosterone replacement therapy might be considered an option to induce neural regeneration and preserve muscle strength in patients with SCI [117]. Notably, men with incomplete SCI treated monthly with testosterone cypionate (200 mg) showed higher American Spinal Injury Association (ASIA) discharge motor scores. However, these results should be interpreted cautiously, as the groups were not randomized and differed by ethnicity and length of stay [119]. Prospective studies are needed to validate these findings.

## 5. Conclusions

Although the pathophysiology of SCI is highly complex, evidence supports the neuroprotective role of sex steroid hormones. Several mechanisms have been proposed by which particular pathways related to neuroprotection may lead to therapeutic targets. E2- and P4-mediated neuroprotection is related to the interactions with their receptors and signaling systems, effects on astrocytes and microglia, modulation of the inflammatory response, and their antioxidant effects. Conversely, testosterone-mediated neuroprotection is more controversial since contrasting results can be found depending on the animal model or patients analyzed. Therefore, it is imperative to develop further studies in experimental models to elucidate the neuroprotective effect of testosterone on SCI.

The optimal doses and duration of hormone therapy in humans are currently unknown due to limited clinical studies. Although E2, P4, and testosterone treatments are generally considered safe, multicenter, randomized, prospective, and larger human clinical studies s are needed to further evaluate their clinical role as neuroprotective agents after acute traumatic spinal cord injury.

## Figures and Tables

**Figure 1 biomedicines-12-01478-f001:**
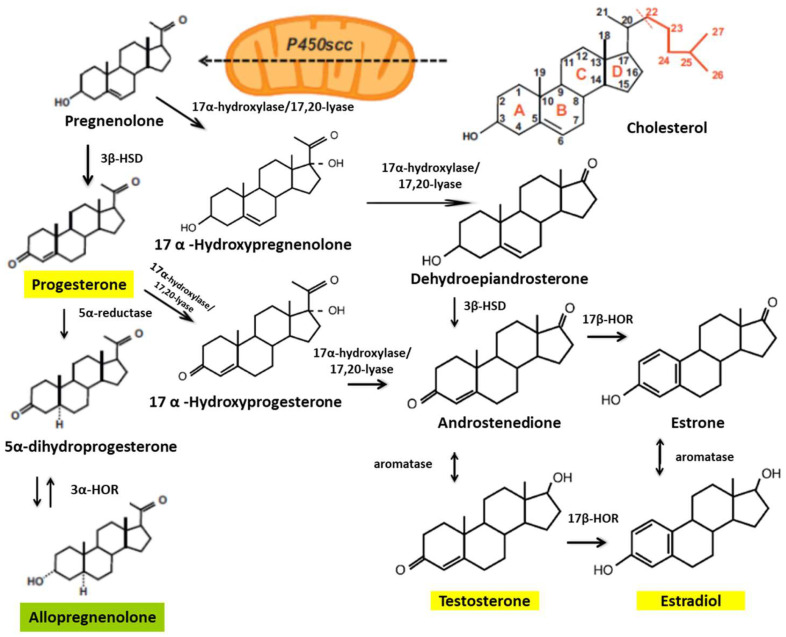
Biosynthetic pathways of steroid hormones. The cytochrome mitochondrial cytochrome P450scc catalyzes the conversion of cholesterol to pregnenolone. Pregnenolone is then converted to progesterone by 3β-hydroxysteroid dehydrogenase (3β-HSD). Pregnenolone can also be converted to hydroxypregnenolone and progesterone to androstenedione by the cytochrome P450c17 (17-hydroxylase/17,20-lyase). The 17β-hydroxysteroid oxidoreductases (17β-HORs, also called 17β hydroxysteroid dehydrogenase) catalyze the final steps of androgen and estrogen biosynthesis (androstenedione ↔ testosterone and estrone ↔ estradiol). 5α-reductase is involved in the metabolism of progesterone, generating 5α-dihydro-progesterone, and 3α-hydroxysteroid oxidoreductase (3α-HOR) metabolizes 5α-dihydro-progesterone to allopregnanolone (modified from Schumacher et al., 2003 [41]).

**Figure 2 biomedicines-12-01478-f002:**
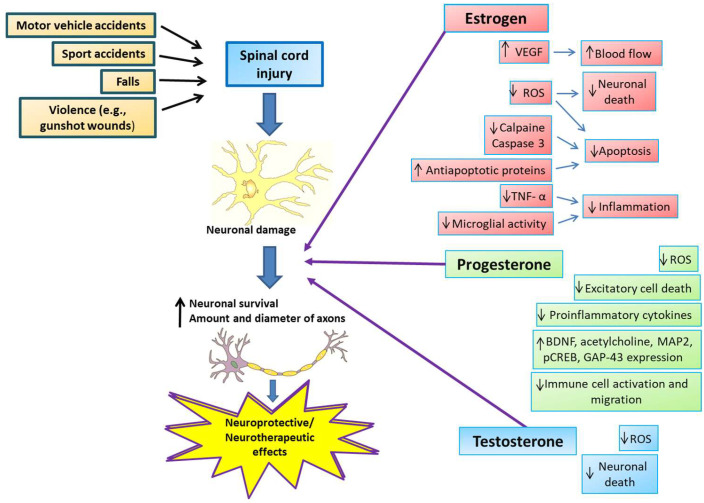
Neuroprotective and neurotherapeutic effects of estrogen, progesterone, and testosterone on spinal cord injury. SCI causes include motor vehicle and sports accidents, falls, and violent encounters, producing neuronal damage directly from the injury and a subsequent secondary cascade of events in the spinal cord. After SCI, estrogen administration reduces tumor necrosis factor-α (TNF-α), microglial activity, and calpain and caspase-3 levels, increases anti-apoptotic protein levels, and increases vascular endothelial growth factor (VEGF) expression, consequently attenuating inflammation, decreasing apoptosis, and increasing blood flow to the site of injury, resulting in a more significant number of axons of larger diameter. Progesterone exerts its neuroprotective actions by increasing brain-derived neurotrophic factor (BDNF), acetylcholine, and other proteins necessary to restore motor function, such as microtubule-associated protein 2 (MAP2), cyclic-AMP response element-binding protein (CREB), and growth-associated protein 43 (GAP-43). Also, progesterone inhibits pro-inflammatory cytokine release and immune cell activation or migration. In addition, progesterone shows antioxidant properties and inhibits excitotoxic cell death. Testosterone administration reduces oxidative stress and cell death, promoting neurological recovery and neural regeneration.

## Data Availability

Not applicable.

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
