# Peer review of "Evaluating Sex Steroid Hormone Neuroprotection in Spinal Cord Injury in Animal Models: Is It Promising in the Clinic?"

_biomedicines, 2024, doi:10.3390/biomedicines12071478_

Round 1
Reviewer 1 Report
Comments and Suggestions for Authors
Thank you for the opportunity to review this paper. The review summarizes the current status of sex steroid hormone in the treatment of spinal cord injury from basic and clinical aspects. Overall, the manuscript structure is well organized and the evidence demonstrated is adequate. However, there are a few concerns that need to be addressed.
1> In terms of a review, the authors seem to list and describe all the study findings one by one. What are the MAIN ideas that the authors want to share with readers in each section? I recommend that authors, in their revision, form their own views first, and then cite and introduce previous research where necessary, rather than piling up all the evidence together.
2> Some sentences in the manuscript are hard to understand and sometimes confusing. For instance, in the Abstract section, "Neuroprotection of these hormones has been associated with ... their antioxidant effects"; In the Highlights section, "Progesterone may be therapeutic targets for spinal cord injury"; In the Introduction, "Depending on the severity of the injury, SCI often leads to death". There could be more of these issues and I suggest the authors go through the manuscript carefully to make it more clear and scientific.
3> Figure 1 is irrelevant to the paragraph where it is quoted? Figure 2, which shows the mechanisms of sex steroid hormone, is useful. However, I suggest the authors include the figure legend in the main text and specify how the drugs work.
4> The contents of tables 1-3 seem tedious. The authors need to simplify what they want to demonstrate.
Comments on the Quality of English LanguageNone
Author Response
Reviewer comments, author responses, and manuscript changes
We thank the referees for carefully reviewing the manuscript and their opinions regarding its scientific content and presentation. In what follows, the reviewers’ comments are in italics, the author’s responses are in blue, and the changes are highlighted.
Reviewer #1
Thank you for the opportunity to review this paper. The review summarizes the current status of sex steroid hormones in the treatment of spinal cord injury from basic and clinical aspects. Overall, the manuscript structure is well organized, and the evidence is adequate. However, there are a few concerns that need to be addressed.
Response: We greatly appreciate the thorough and thoughtful comments the referee provided on our paper. We are sure that these comments helped improve our manuscript.
Specific comments
- In terms of a review, the authors seem to list and describe all the study findings one by one. What are the MAIN ideas that the authors want to share with readers in each section? I recommend that authors, in their revision, form their own views first, and then cite and introduce previous research where necessary, rather than piling up all the evidence together.
Response: We appreciate this suggestion. Throughout the text, we highlighted the main ideas we wanted to share from the articles on which we based our review. We then detailed the information with the results of previous research.
- Some sentences in the manuscript are hard to understand and sometimes confusing. For instance, in the Abstract section, "Neuroprotection of these hormones has been associated with ... their antioxidant effects"; In the Highlights section, "Progesterone may be therapeutic targets for spinal cord injury"; In the Introduction, "Depending on the severity of the injury, SCI often leads to death". There could be more of these issues and I suggest the authors go through the manuscript carefully to make it more clear and scientific.
Response: Thank you for these language corrections. We revised the entire manuscript to improve the language according to the journal's standards. In addition, we have replaced these phrases as follows:
As neuroprotection has been associated with signaling pathways, the effects of these hormones are observed on astrocytes and microglia, modulating the inflammatory response, cerebral blood flow, and metabolism, mediating glutamate excitotoxicity, and their antioxidant effects. (Abstract, page 1, lines 37-39).
Progesterone, estradiol, and testosterone may be used as therapeutic agents for spinal cord injury (Highlights, page 1, lines 47)
Depending on the severity of the injury, complications often result in the death of SCI patients (Introduction section, page 2, lines 53-54)
- Figure 1 is irrelevant to the paragraph where it is quoted? Figure 2, which shows the mechanisms of sex steroid hormones, is useful. However, I suggest the authors include the figure legend in the main text and specify how the drugs work.
Response: Thank you for your observations. We have quoted both figures accordingly as follows:
Some studies have shown that treatments with E2, P4, and testosterone—synthesized from cholesterol—reduce the damage caused by SCI (Figure 1). (Sex Steroid Hormones section, page 3, lines 123-124).
In addition, SSH treatments are relatively accessible and inexpensive, which enhances their potential for widespread use [50] (Figure 2). (Sex Steroid Hormones section, page 3, lines 130-132).
For Figure 2, we consider the explanation of the mechanisms of action in the figure legend a summary. For this reason, we have preferred to leave them as figure legends since these mechanisms of action of sex steroid hormones were detailed throughout the text.
- The contents of tables 1-3 seem tedious. The authors need to simplify what they want to demonstrate.
Response: Thank you for your observation. We revised the tables and tried simplifying them at the reviewer's suggestion. However, the tables show a compilation of the results reported in the literature. For this reason, we decided to keep them since we hope that our contribution to compiling this information will be a reference for future researchers.

Reviewer 2 Report
Comments and Suggestions for Authors
The manuscript evaluates the potential sex steroid hormone (SSH) as neuroprotective agents for spinal cord injury (SCI) treatment at the experimental level. In this manuscript, the authors stated that early administration of SSH can mitigate the pathophysiological mechanisms and effects of SCI. The authors focused on reviewing three main types of SSH: estradiol, progesterone, and testosterone, to evaluate their potential for neurological recovery and decreasing mortality rates. Animals were the main target of studies in this manuscript to assess the effects of SSH on SCI recovery, while only a few studies on humans, mainly focusing on testosterone, are included. The authors concluded that estradiol and progesterone can mediate neuroprotection in SCI animals through interactions with their receptors and signaling systems to modulate inflammatory response and antioxidant effects. Nevertheless, the neuroprotective effect of testosterone remains controversial in animal models and requires more experimental studies to evaluate its potential as neuroprotective agents. As mentioned, SSH therapeutic studies in humans still need further research, especially at the clinical level. The content of the manuscript was well-organized and easy to follow. However, a few minor questions need to be addressed before publication.
1. The authors should clarify that the review mainly focuses on animal models while only discussing testosterone in humans due to lack of studies in abstract. This can inform the readers that the review was done on animal models with a lack of study in humans.
The authors should write the non-abbreviated term or full term when it first appears in the manuscript or before using the abbreviation in the manuscript. Please see the example in this article: "PGRMC1 effects on metabolism, genomic mutation and CpG methylation imply crucial roles in animal biology and disease." Please check the other abbreviations, such as E2, P4, PGRMC1, VEGF, MAP2, HRT, and others, are written in full term before the abbreviated terms are being used in the manuscript.
Author Response
Reviewer’s comments, author responses, and manuscript changes
We thank the referees for carefully reviewing the manuscript and their opinions regarding its scientific content and presentation. In what follows, the reviewers’ comments are in italics, the author’s responses are in blue, and the changes are highlighted.
Reviewer #2
The manuscript evaluates the potential sex steroid hormones (SSHs) as neuroprotective agents for spinal cord injury (SCI) treatment at the experimental level. In this manuscript, the authors stated that early administration of SSH can mitigate the pathophysiological mechanisms and effects of SCI. The authors focused on reviewing three main types of SSH, estradiol, progesterone, and testosterone, to evaluate their potential for neurological recovery and decreasing mortality rates. Animals were the main target of studies in this manuscript to assess the effects of SSH on SCI recovery, while only a few studies on humans, mainly focusing on testosterone, are included. The authors concluded that estradiol and progesterone can mediate neuroprotection in SCI animals through interactions with their receptors and signaling systems to modulate inflammatory response and antioxidant effects. Nevertheless, the neuroprotective effect of testosterone remains controversial in animal models and requires more experimental studies to evaluate its potential as a neuroprotective agent. As mentioned, SSH therapeutic studies in humans still need further research, especially at the clinical level. The content of the manuscript was well-organized and easy to follow. However, a few minor questions need to be addressed before publication.
Response: We are deeply grateful for the suggestions provided by the reviewer to improve our work.
- The authors should clarify that the review mainly focuses on animal models while only discussing testosterone in humans due to the lack of studies in the abstract. This can inform the readers that the review was done on animal models with a lack of studies in humans.
Response: Thank you for your suggestion. We agree with the reviewer that our review mainly focuses on animal models. Therefore, we specified this information in the title, abstract, and the aim of the review as follows:
Title:
Evaluating sex steroid hormone neuroprotection in spinal cord injury in animal models: Is it promising in the clinic? (page 1, line 2)
This review evaluated evidence supporting hormone-related neuroprotection over SCI and the possible underlying mechanisms in animal models. (Abstract, page 1, lines 35-36).
This review aims to analyze the evidence in studies that support hormone-related neuroprotection on SCI and the possible underlying mechanisms in animal models. (Introduction section, page 2, lines 71-72)
- The authors should write the non-abbreviated term or full term when it first appears in the manuscript or before using the abbreviation in the manuscript. Please see the example in this article: "PGRMC1 effects on metabolism, genomic mutation, and CpG methylation imply crucial roles in animal biology and disease." Please check the other abbreviations, such as E2, P4, PGRMC1, VEGF, MAP2, HRT, and others, are written in full terms before the abbreviated terms are used in the manuscript.
Response: Thank you for your observation. We revised the whole manuscript to include the non-abbreviated or full term when it appeared for the first time and corrected all typos and omissions. Corrections throughout the manuscript are highlighted in yellow.
